# Health Knowledge of Lifestyle-Related Risks during Pregnancy: A Cross-Sectional Study of Pregnant Women in Germany

**DOI:** 10.3390/ijerph17228626

**Published:** 2020-11-20

**Authors:** Anja Oechsle, Michel Wensing, Charlotte Ullrich, Manuela Bombana

**Affiliations:** 1Department of General Practice and Health Service Research, University Hospital Heidelberg, Im Neuenheimer Feld 130.3, 69120 Heidelberg, Germany; anja.oechsle@googlemail.com (A.O.); michel.wensing@med.uni-heidelberg.de (M.W.); charlotte.ullrich@med.uni-heidelberg.de (C.U.); 2Department of Health Promotion, AOK Baden-Württemberg, Presselstrasse 19, 70191 Stuttgart, Germany

**Keywords:** prenatal care, knowledge, pregnancy, lifestyle, risk factors, alcohol, smoking, nutrition, attitudes

## Abstract

This study aimed to investigate (1) pregnant women’s level of knowledge of lifestyle-related risk factors during pregnancy and their potential health impact on their offspring, and (2) the factors affecting women’s knowledge of lifestyle-related risk factors during pregnancy. A cross-sectional observational study of pregnant women was carried out in obstetric and gynecologic care settings at three hospitals in Southern Germany. Data from 209 pregnant women revealed large knowledge gaps on lifestyle-related risk factors during pregnancy and their potential health impact. Factors affecting women’s knowledge of lifestyle-related risk factors during pregnancy were specifically associated with socioeconomic status, e.g., lower household net income, middle educational level, and statutory health insurance status. Women who had received information from their gynecologist had a higher level of knowledge of lifestyle-related risk factors during pregnancy. This study showed that health promotion regarding lifestyle-related risks during pregnancy specifically needs to address women from the low-to-middle socioeconomic status group. Gynecologists seem particularly effective in providing this information.

## 1. Background

Maternal lifestyle-related health behaviors during pregnancy impact their offspring’s health status. These adverse health behaviors cover issues on alcohol consumption, smoking, coffee consumption, nutrition, food supplements, medication, oral health, and physical activity. However, a major condition of healthy behaviors is the ability to understand health information and how to access it. The conditions for healthy behavior are understanding the multi-facetted demands of heath, health recommendations, and instructions in our society, and adapting health behaviors accordingly [1]. A recent study by Schaeffer et al. showed that, currently, 44% of those surveyed over the age of 15 have clear difficulties in understanding and assessing health information [2].

### 1.1. Alcohol Consumption

Prenatal alcohol exposure (PAE) is associated with a variety of fetal malformations, including cognitive deficits, behavioral abnormalities and growth disorders, as captured by the umbrella term fetal alcohol spectrum disorder (FASD) [3,4,5]. Recommendations on alcohol consumption during pregnancy suggest complete avoidance. However, many women still drink alcohol during pregnancy; in Germany, 14.0% reported drinking alcohol occasionally during pregnancy [6]. The prevalence rates of maternal alcohol intake during pregnancy in Europe range from 4.1% in Norway to 28.5% in the U.K. [7]. A meta-analysis from 2017 reported data on the prevalence of FASD: 111.1 per 1000 births in South Africa, 19.8 per 1000 births in Europe, and 53.3 per 1000 births in Croatia. The lowest prevalence was estimated in the World Health Organization (WHO) eastern Mediterranean region, with 0.1 per 1000 births [3].

### 1.2. Smoking

Smoking during pregnancy can have severe health effects on offspring and it is recommended to completely abstain from smoking during pregnancy [8,9,10]. A meta-analysis demonstrated that prenatal smoke exposure (PSE) increases the risk of stillbirths by 47.0% [8]. In another meta-analysis, the authors reported that PSE is associated with preterm birth (PTB) and growth disorders, especially unusual head size and femur length [11]. Further studies have shown that PSE is associated with asthmatic diseases, negative epigenetic effects and gen deregulations, weight problems, and low birth weight (LBW), which is associated with hypertension in adulthood [11,12]. Abstinence or a reduction in smoking during pregnancy can reduce growth disturbances [10]. The prevalence of smoking during pregnancy has been stated to be 8.1% (95% confidence interval (CI), 4.0–12.2) in Europe, which is much higher than the worldwide value of 1.7% (95% CI, 0.0–4.5). In terms of European countries, Ireland has the highest rate, with 38.4% (95% CI, 25.4–52.4), followed by Bulgaria with 29.4% (95% CI, 26.6–32.2) [13]. In Germany, the KiGGS Study (German Health Interview and Examination Survey for Children and Adolescents) showed that 12.1% of women smoked during pregnancy [6].

### 1.3. Coffee Consumption

Dietary habits and nutrition during pregnancy are closely related to maternal, embryonic, and fetal health statuses, as a range of risks can be avoided by appropriate gestational nutrition [14,15,16]. According to the German maternity guidelines [17], coffee consumption may be a risk factor during pregnancy. According to the current evidence base, as derived from a meta-analysis, amounts above 150 mg per day (approximately one cup) are associated with higher risks of LBW [18]. Results from another meta-analysis suggested a dose–response relationship between coffee consumption and the risk of pregnancy loss [19]. A recent study from Korea showed that even less than one cup of coffee (light coffee drinker) and one cup of coffee per day (moderate coffee drinker) during pregnancy are associated with a significantly increased risk of bleeding in early pregnancy [20]. Coffee intake above one cup per day during pregnancy was found to be associated with childhood acute myeloid leukemia [21].

### 1.4. Nutrition

The recommendations from the German nutrition society on a healthy and balanced diet during pregnancy include daily consumption of certain food groups, such as vegetables, fruits, whole grains, low-fat milk, and sufficient amounts of water [22]. Studies have shown that the benefits of these food groups are evident [23,24]. The weekly consumption of oily fish and low-fat meat is recommended as an appropriate diet [22,25]. Fats and oils should be consumed at a less-frequent level, and fats should be high in quality. As recommended by the American Heart Association, the consumption of unsaturated fats should be favored over the consumption of saturated fats. The consumption of unsaturated fatty acids contributes to the prevention of heart diseases and provides additional vitamins for the mother and the offspring. The consumption of sweets and soft drinks should be avoided as they contain large amounts of saturated fats and sugars [23,26]. The WHO recommends consuming a maximum of 10.0% of one’s daily energy intake in the form of free sugars [22,27,28], which means a maximum amount of 50 g. In order to achieve health benefits, this amount should be halved [29]; this also applies to pregnant women [22,27,28]. Complete avoidance of specific foods during pregnancy, such as raw meat, raw fish, soft cheese, or energy drinks, is essential in the prevention of human toxoplasmosis or listeria infections.

The maternal energy requirement increases in the third trimester by approximately 10.0% [14]. However, inadequate gestational weight gain is associated with a range of adverse health effects in offspring, such as diabetes, attention-deficit hyperactivity disorder, hypertension, and asthma [6,30]. The recommendation for maximum weight gain depends on the body mass index (BMI) of the pregnant women. The recommended weight gains are 12.5–18.0 kg for underweight women (BMI < 18.5), 11.5–16.0 kg for women of normal weight (BMI 18.5–24.9), 7.0–11.5 kg for overweight women (BMI 25–29.9), and 5.0–9.0 kg for obese women (BMI > 30) [30]. 

Although many studies have investigated the complete avoidance of specific food categories during pregnancy [22], annually, there are still 6393 seroconversions and 1279 fetal infections with Toxoplasma gondii in Germany [31]. Since 2009, there has been a significant increase in the number of cases of listeriosis, from 396 cases in 2009 to 771 cases in 2017. Pregnant women with listeriosis infection have an increased risk of PTB or miscarriage, and fetal listeriosis infection is associated with an increased risk of mortality [32]. Ten percent of listeriosis cases are pregnancy-associated, affecting both the mother and the offspring [33]. The risk of infection during pregnancy is 18 times higher compared to the general population [34]. 

Vegan dietary habits during pregnancy are associated with fetal growth restrictions, developmental disorders, severe damage of the nervous system, and pre-eclampsia [35,36]. For vegan women, individual nutrient consultations and regular monitoring during pregnancy are relevant to prevent deficiency symptoms in their offspring [22].

### 1.5. Food Supplements

Pregnant women with a vegan diet have a 10.0% increased need for proteins, and should therefore favor omega 3 fatty acids and monounsaturated oils, and avoid trans fats. Plant-based foods containing high levels of calcium, vitamin D, and vitamin B12 supplements are essential for preventing nervous system birth defects [37].

During pregnancy, the need for nutritional supplements, such as folic acid (300–550 µg) and iodine (200–230 µg) [38], is significantly increased [22]. A folic acid deficiency can result in severe health impairments, such as pre-eclampsia [39]; to reduce the risk of pre-eclampsia, pregnant women with a low intake of calcium or women under 19 years of age should supplement calcium by 1000–1200 mg [40]. Iron supplementation depends on the individual’s iron status [36]. According to the latest recommendations, all nutrients, with the exception of iodine and folic acid, are generally covered by a balanced diet [22,36,37,38,39]. 

### 1.6. Physical Activity

Physical inactivity in combination with high energy intake during pregnancy is associated with maternal overweight and obesity, which, in turn, might result in diabetes for the mother, later childhood overweight and obesity, and many other undesirable health outcomes [14,41]. A meta-analysis suggested that physical activity significantly reduces the likelihood of a cesarean section [41], due to lower weight gains, resulting in a decreased diabetes risk, which is often accompanied by birth complications. Moreover, physically active pregnant women show less symptoms of depression [42]. Therefore, the guidelines from Canada, the U.K., and Denmark recommend moderate exercise of 30 min per day, two to three times a week during pregnancy, based on individual fitness level. However, recommendations vary across countries [42]. Essentially, sports with a low risk of injury, such as Nordic walking, swimming, or yoga, are usually recommended during pregnancy. High-impact activities, such as diving, contact sports, bodybuilding and height sports, should be renounced [42,43,44,45].

### 1.7. Oral Health

The German maternity guidelines, as established by the Federal Joint Committee (Gemeinsamer Bundesausschuss, G-BA), request the informing of pregnant woman on oral health during pregnancy, especially in light of periodontitis prevention [17]. A recent systematic review demonstrated an association between gestational periodontitis and PTB and LBW [46].

### 1.8. Medications

Medications during pregnancy can harm the embryo and fetus, as substances can cross the placenta and thus can directly reach the fetal blood circulation. It is recommended to consult a gynecologist before taking any medication during pregnancy [12,29].

In order to prevent lifestyle-related risks and diseases during pregnancy, it is necessary to inform women on adequate health behaviors during pregnancy. In order to draw a picture of the need for an intervention on health behaviors during pregnancy and to derive policy recommendations for gynecological care, this study aimed to investigate: (1) pregnant women’s level of knowledge of lifestyle-related risk factors during pregnancy and their potential health impact on their offspring, and (2) which factors affect women’s knowledge of lifestyle-related risk factors during pregnancy.

## 2. Methods

### 2.1. Study Design and Setting

This study was a cross-sectional observational study of pregnant women’s knowledge of lifestyle-related risks during pregnancy, conducted in three antenatal hospitals in southern Germany between September and October 2018. Participants were purposefully recruited from different types of hospitals: A large care public hospital, 1559 beds, level 4 (maximum care), 3400 births per year;A Christian hospital, 580 beds, level 3 (central care level), 1000 births per year;A small public hospital, 277 beds, level 2 (basic and standard care), 800 births per year.

Of the six contacted hospitals, three agreed to participate in the study. 

### 2.2. Study Population and Data Collection

Pregnant women of legal age (≥18 years of age), able to read and understand the German language (12.1% of the working age population cannot, or only to a very limited level, read and write; 17.6% of non-Germans living in Germany have difficulties in understanding German [47]), who attended the delivery room management or the pregnant outpatient clinic during the survey period were eligible to participate in our study. A total of 260 women were randomly approached and selected during their waiting time for the delivery room management or in the pregnant outpatient clinic. The women were addressed individually and were invited to participate in the study. Three women did not meet the inclusion criteria and 48 women refused to participate. A total of 209 pregnant women filled in the questionnaire (response rate: 80.4%), of whom 88.5% (*n* = 185) were recruited before the start of the delivery room management and 11.5% (*n* = 24) during their waiting period in the pregnancy outpatient clinic.

Data were collected anonymously to reduce potential social desirability bias. Informed consent was obtained by inserting the questionnaire into a box. The dataset is provided by the corresponding author on reasonable request.

### 2.3. Questionnaire

A literature search did not identify a German validated questionnaire on the topic of interest, and therefore, we developed a quantitative questionnaire (available on request from the corresponding author).

The German questionnaire consisted of 22 items. Multiple choice questions with single-select or multi-select answer options were applied. The items covered topics on sociodemographic features and the knowledge and attitude of pregnant women with respect to alcohol consumption, smoking, nutrition and supplementation, physical activity, oral health, and medication. The questionnaire was pre-tested by three pregnant women and adjustments were made accordingly. The women completed the questionnaire within 9–11 min. 

### 2.4. Dependent Variables

The dependent variables were women’s knowledge of lifestyle-related behaviors during pregnancy, including knowledge of alcohol consumption, smoking, coffee consumption, recommended nutrition, avoidable nutrition, supplementation, medication, oral health, kind of physical activity, and intensity of physical activity. Three further dependent variables were also included, namely, pregnant women’s knowledge of the health effects of alcohol consumption, smoking, and a vegan diet.

All variables were categorized into dichotomous outcome variables (i.e., consistent or inconsistent with recommendations). Each variable was based on a set of multiple answer categories, for which the women had to select single or multiple correct answers. “Correct answers” were based on existing federal guidelines and recommendations [14,17,23,38,48]. For example, women’s knowledge of alcohol consumption was measured by the question “What do you think on how many glasses of alcohol during pregnancy are harmless?” Women had to insert the number of glasses per day/week/month; “0” was the correct number and answer. To assess women’s knowledge of the health effects of alcohol consumption, we asked the question “What do you think which effects may be caused by alcohol consumption during pregnancy?” The possible answers were: “Moderate consumption has positive health effects during pregnancy”; “frequent consumption has no health effects on the mother and the unborn child”; “alcohol during pregnancy, regardless of its quantity, may have adverse health effects on the unborn child across their lifespan”; “even small amounts of alcohol consumption during pregnancy may have severe effects, such as fetal alcohol syndrome”; “only regular alcohol consumption may cause fetal alcohol spectrum disorders and fetal alcohol syndrome.” The correct answers were: “Alcohol during pregnancy, regardless of its quantity, may have health effects on the unborn child across their lifespan” and “even small amounts of alcohol consumption during pregnancy may have severe effects, such as fetal alcohol syndrome.”

### 2.5. Independent Variables

The independent variables were maternal age (continuous), week of pregnancy (continuous), number of previous pregnancies (continuous), number of biological children (continuous), level of education (categorial: low/middle/high), household net income (categorial: low/middle/high), marital status (categorial: in a partnership/not in a partnership), insurance status (categorial: statutory health insurance/private health insurance), and information received by a gynecologist (categorical: yes/no). 

Educational level was classified into low (secondary school and high school degree), middle (qualifying degree for German universities and one-year junior college level/A-level), and high (academic degree and doctoral degree). Household net income was classified into low (EUR ≤ 2.500), middle (EUR 2.501–4.000), and high (EUR ≥ 4.001) [49].

### 2.6. Data Analysis 

Data were analyzed descriptively. We used absolute frequencies, percentages, mean values, *p*-values for all differences (*p* < 0.05), and standard deviations. Multivariate logistic regression analyses were applied to investigate the likelihood of giving “wrong answers” (i.e., answers that contrasted with prevailing scientific knowledge) to questions concerning the effects of alcohol consumption, smoking, coffee, nutrition, avoidable nutrition, supplements, medication, oral health, kind of physical activity, intensity of physical activity, and a vegan diet. We declared a value as a mistake as soon as the item did not correspond to the current recommendations in the literature. All tests were conducted for 95% confidence with α = 0.05. Data were analyzed using IBM SPSS Statistics for Windows, Version 25.0 (IMB Corp. Released 2017. Armonk, NY, USA: IBM).

Ethical approval for this study was obtained from the Ethics Committee of the Medical Faculty of the Ruprecht Karls University Heidelberg (S-338/2018). This study was performed in compliance with the Declaration of Helsinki [50]. Participants were informed that by completing and delivering the questionnaire, their approval to participate in this study would be confirmed.

## 3. Results 

### 3.1. Level of Knowledge of Lifestyle-Related Risk Factors and Their Potential Health Effects on the Offspring

Table 1 provides an initial overview of the characteristics of the study sample. It should be noted that 54.4% of the surveyed women had a high education level and 53.3% had a high household net income; therefore, these groups were overrepresented.

Table 2 is a frequency table on pregnant women’s level of knowledge of lifestyle-related risk factors during pregnancy and their potential health impact on their offspring. Table 3 is a frequency table on pregnant women’s knowledge of the health effects of alcohol consumption, smoking, and a vegan diet.

The analyses demonstrated large knowledge gaps concerning lifestyle-related risks during pregnancy. Although 97.1% of the surveyed women knew that one should abstain from alcohol during pregnancy, 31.8% did not know the specific adverse health effects of alcohol consumption. Meanwhile, 10.8% believed that only regular alcohol consumption causes FASD, and 77.9% of the pregnant women knew that even small amounts of alcohol may have health effects on the unborn child. Occasional consumption of champagne during pregnancy was considered harmless by 3.6% of the women, while 87.7% knew that PAE, regardless of the amount of alcohol, may cause health problems across the lifespan. Similarly, 99.0% of the women knew that one should not smoke during pregnancy, although 48.5% did not know about the adverse effects of smoking. Meanwhile, 29.1% of the women did not believe that smoking may result in life-long asthmatic diseases in their offspring, 29.6% did not believe that smoking can lead to a miscarriage, 12.6% did not believe that smoking can cause health effects in both the mother and the offspring, 11.2% did not believe that smoking can cause growth restrictions in the embryo and fetus, and 7.3% did not believe that smoking can cause developmental delays. Moreover, 82.3% did not know about the adverse effects of a vegan diet. Of the pregnant women, only 21.2% correctly assessed that a vegan diet without supplementation is a risk for rickets (softening and deformations of the bones) in offspring, while 6.9% of the women believed that vegan nutrition has neither positive nor negative health effects on the embryo and fetus. Furthermore, 2.5% of the study participants considered a vegan diet to positively affect maternal and infant health during pregnancy.

Of the surveyed women, 70.8% did not know about the contents of recommended nutrition and 54.5% did not know about avoidable nutrition. Moreover, 63.0% believed that more than one cup of coffee per day is safe, while 37.0% agreed with the recommendations on not consuming more than one cup of coffee per day. However, 93.2% did not know the recommended supplementations during pregnancy. Concerning the intensity of physical activity, 98.6% of the participants agreed with the statement that a pregnant woman should perform moderate physical activity for at least 20 min per day. However, 48.1% did not know what kind of physical activity is recommended during pregnancy. On the question of how to deal with influenza and medication during pregnancy, 99.5% answered correctly that an appointment with a physician is necessary. Concerning oral health, 96.6% of the participants were aware that teeth are more susceptible to tooth decay and gingivitis during pregnancy.

### 3.2. Factors Affecting Women’s Knowledge of Lifestyle-Related Risk Factors during Pregnancy

We investigated the risks of health knowledge gaps with respect to the general recommendations on lifestyle-related risk factors during pregnancy in association with independent variables, as shown in Table 4. We selected outcomes with a high rate of non-compliant answers with respect to the general recommendations: coffee consumption, recommended nutrition, avoidable nutrition, supplementation, and kind of physical activity.

Women with a middle educational level had a significantly lower risk of being non-compliant in their answers to the general recommendations on coffee question, as compared to woman with a higher educational level (odds ratio (OR), 0.42; 95% CI, 0.19–0.90). However, they had a significantly higher risk of being non-compliant in their answers concerning the general recommendations on physical activities during pregnancy (OR, 2.78; 95% CI, 1.30–5.98).

Pregnant woman with a lower household net income level were at a significantly higher risk of being non-compliant in their answers concerning the general recommendations on avoidable nutrition as compared to women with a higher household net income level (OR, 7.45; 95% CI, 2.59–21.42)

Sensitivity analyses showed that the women who received information on lifestyle-related risks during pregnancy from a gynecologist were at a significantly lower risk of being non-compliant concerning the general recommendations on medication intake during pregnancy (OR, 7.71; 95% CI, 1.00–59.73).

Meanwhile, the risk of non-compliant knowledge of the health effects of alcohol consumption, smoking, and a vegan diet during pregnancy, in association with independent variables, is shown in Table 5.

Pregnant woman with a lower household net income level were at significantly higher risk of being non-compliant in their answers concerning the state of the art on the health effects of alcohol consumption (OR, 2.82; 95% CI, 1.14–7.03), smoking (OR, 2.31; 95% CI, 0.98–5.46), and a vegan diet (OR, 3.93; 95% CI, 1.00–15.37) during pregnancy, as compared to pregnant women with a higher household net income level.

Pregnant women with statutory health insurance were also at a significantly higher risk of being non-compliant in their answers concerning the state of the art on the health effects of alcohol consumption during pregnancy as compared to women with private health insurance (OR, 5.06; 95% CI, 1.11–22.98).

## 4. Discussion

### 4.1. Key Findings

Our results from a cross-sectional study on 209 pregnant women demonstrated large knowledge gaps on lifestyle-related risk factors during pregnancy and their potential health impact. We identified substantially large gaps in women’s knowledge of coffee consumption, recommended and avoidable nutrition, and supplementation during pregnancy. The women’s knowledge of oral health, alcohol consumption, and smoking during pregnancy was high; however, they underestimated and falsely estimated the potential health effects of alcohol consumption, smoking, and a vegan diet during pregnancy. Most estimation mistakes were related to knowledge of recommended supplementation during pregnancy.

We identified factors affecting women’s knowledge of lifestyle-related risk factors during pregnancy, although these findings should not be overinterpreted. These factors were specifically associated with the women’s socioeconomic status. Women with a lower household net income, a middle educational level (does not apply to coffee consumption), and statutory health insurance status were at higher risk of falsely estimating and underestimating lifestyle-related risk factors during pregnancy. However, women with a middle educational level had a lower risk of knowledge gaps on recommended levels of coffee consumption during pregnancy, as compared to higher educated women.

### 4.2. Discussion of the Key Findings

#### 4.2.1. Pregnant Women’s Level of Knowledge of Lifestyle-Related Risk Factors during Pregnancy and Their Potential Health Effects on Their Offspring

Of the surveyed women, 63.0% estimated more than one cup of coffee per day during pregnancy as being harmless. Guidelines across the world are inconsistent in their recommendations regarding the amount of coffee intake during pregnancy [20,21,22,51,52]. Thus, the limited knowledge base on the safe amounts of coffee consumption during pregnancy may be a result of limited and inconsistent guidelines and recommendations [52,53,54,55,56]. Of the surveyed women, 70.8% were not able to correctly select recommended food groups in terms of daily nutrition, such as vegetables, fruits, and unsweetened drinks (tea and water), and instead selected food groups such as sweets and sugar-sweetened beverages. Moreover, 54.5% of the pregnant women were not able to correctly select food groups that should be avoided during pregnancy, according to the recommendations from the German Society of Nutrition (DGE), such as raw eggs, raw meat, and raw sausage (e.g., salami) [22]. We found the largest knowledge gaps in the question on nutritional supplementation during pregnancy: 93.2% of the pregnant women did not know what one should supplement during pregnancy (i.e., folic acid and iodine), or not supplement if not specifically recommended by the doctor (such as vitamin H and omega-3 fatty acids), as recommended by DGE. However, 58.0% correctly answered that iodine is a recommended nutritional supplement during pregnancy, and 98.1% answered that folic acid is a recommended nutritional supplement during pregnancy. Our results are in line with those of Lee et al., reporting that women have limited information about dietary guidelines for healthy eating [57]. In a study from Singapore, only 56.4% of the surveyed women knew about the increased need for folic acid during pregnancy [58].

We also investigated women’s knowledge of the health effects of alcohol consumption, smoking, and a vegan diet during pregnancy. Almost all of the surveyed women knew that abstinence from alcohol and smoking is recommended during pregnancy; however, a substantial proportion of the surveyed women did not know the potential effects of alcohol consumption and smoking during pregnancy. These results are similar to those from a focus group study, suggesting that women have heard the name of some effects already but have a lack of information on the effects of maternal alcohol intake during pregnancy [59]. In a German study from 2017, 30% of the surveyed women did not believe that alcohol consumption during pregnancy can cause adverse lifelong health outcomes [60]. In this 2017 study, only 17% of the pregnant women were informed by their gynecologists about the avoidance of alcohol consumption during pregnancy, and 38% of the surveyed women had not found any information in the media on alcohol consumption during pregnancy [60].

In our study, 48.5% did not correctly assess the effects of smoking during pregnancy. Similar results were found in the HealthStyles study from 2008, conducted in the U.S., demonstrating that only 23% of the surveyed women of reproductive age had high knowledge of the adverse health effects of smoking during pregnancy [61]. In our study, 70.4% of the women believed that smoking can cause miscarriage. In the HealthStyles study, 72.9% of women of reproductive age believed that smoking can cause miscarriage. Similarly, in our study, 88.8% believed that smoking can cause growth problems in the embryo and fetus, in comparison to the HealthStyles study’s 92.7%, who believed that smoking has potentially adverse effects on fetal growth. In total, the adverse effects of smoking during pregnancy were underestimated and suggest that women’s health knowledge of lifestyle-related risks during pregnancy has the potential to be improved.

Of the surveyed women, 82.3% did not correctly assess the adverse effects of a vegan diet during pregnancy and substantially underestimated its potential health effects. As only a minority of 4.3% of Germans are vegan, it is reasonable that only little is known about the adverse effects of a vegan diet during pregnancy and its potential health effects among the studied population [62]. Several studies on the nutritional knowledge of midwives [63,64,65] and health professionals [66,67,68,69] have demonstrated large knowledge gaps. In the study of Lee et al. [69], women reported receiving limited nutrition advice, and health professionals reported providing limited nutrition advice. The pregnant women’s limited knowledge of the adverse effects of a vegan diet may therefore have resulted from, among others, the limited information provided by and knowledge attained from health professionals.

#### 4.2.2. Factors Affecting Women’s Knowledge of Lifestyle-Related Risk Factors during Pregnancy

The results from our study revealed that the women’s household net income, educational level, and statutory health insurance status were associated with their knowledge of lifestyle-related risk factors during pregnancy.

In previous studies, we found that women with a higher socioeconomic status (measured in terms of household net income, educational level, and job position) were at higher risk of consuming alcohol during pregnancy, but at a lower risk of smoking during pregnancy [70,71,72]. However, our study demonstrated that women with a lower household net income were at significantly higher risk of underestimating and falsely estimating the adverse health effects of alcohol intake, smoking, and a vegan diet during pregnancy on their offspring.

Sensitivity analyses showed that lower household net income levels were significantly associated with higher probabilities of mistakes in answers concerning the effects of alcohol intake, smoking, and a vegan diet during pregnancy. In sum, all of the results together indicate that knowledge of the adverse effects of alcohol intake during pregnancy alone is not enough to cause abstinence from alcohol during pregnancy. Therefore, there might be other confounding factors, moderating and/or mediating the causal chain between knowledge of the adverse effects of alcohol intake during pregnancy and consuming alcohol during pregnancy. Further studies need to investigate these relationships to better understand the mechanisms of alcohol consumption during pregnancy.

In contrast to the results on alcohol consumption, and as expected, women with a lower household net income were at a potentially higher risk of smoking during pregnancy and of underestimating and falsely estimating the adverse health effects of smoking during pregnancy. Similar results were seen in nutritional topics, where higher household net income levels were associated with greater knowledge of the adverse effects of a vegan diet on the offspring and greater knowledge of which foods women should avoid during pregnancy. These results are in line with studies investigating health literacy—also capturing issues of knowledge and understanding of health demands—suggesting that higher socioeconomic status levels are associated with higher levels of health literacy [73,74].

Women with statutory health insurance were at a significantly higher risk of underestimating and falsely estimating the adverse health effects of alcohol intake during pregnancy on their offspring. Sensitivity analyses showed that a larger proportion of women with private health insurance received information on lifestyle-related risk factors from their gynecologist. In contrast, a larger proportion of women with statutory health insurance did not receive information on lifestyle-related risk factors during pregnancy from their gynecologist. Health insurance status might be associated with the time provided by a gynecologist, and thus could result in greater knowledge in people with private health insurance. There was no evidence of any effect of health insurance status on knowledge of the adverse health effects of smoking during pregnancy on the offspring. An explanation might be that among the general population, the effects of smoking during pregnancy might be more well-known than the effects of alcohol consumption during pregnancy [75,76]. Therefore, we believe that the surveyed women with private health insurance might benefit from more intensive supervision, including information on the effects of alcohol consumption during pregnancy, as provided by their gynecologists. Thus, statutory health insurance companies should implement interventions to increase the knowledge of lifestyle-related risk factors during pregnancy.

As a further factor associated with the knowledge of recommended levels of coffee consumption and recommended kinds of physical activities, we identified educational level. Women with middle levels of education were at higher risk of having less knowledge of the recommended kinds of physical activities as compared to higher-educated women. In contrast, women with middle levels of education were at lower risk of having less knowledge of the recommended levels of coffee consumption as compared to higher-educated women. The results from our study suggest that the women were aware of the changes in oral health, knowing that teeth are more susceptible to tooth decay and gum disease during pregnancy. Another German study reported that only one in four women was informed about the necessity of dental check-ups by their gynecologists [77]. However, information on dental health may be transmitted to pregnant women elsewhere and in another context.

### 4.3. Evaluation of Potential Limitations

This study has some limitations. First, our results were based on a small sample size and, thus, our preliminary results need to be interpreted with caution. However, we discussed each result in detail and set, when possible, our results in the context of the current state of the art. Second, if compared to the general population, with 24.4% in the low, 56.0% in the middle, and 19.6% in the high income groups, the high income group was overrepresented in this study. Similarly, in the general population, only 14.1% have a high level of education [78]. We believe that the overrepresentation of women with higher levels of education and income in our study sample is because, specifically, women with higher income and education levels attend non-obligatory delivery room tours. Lower-educated women are supposedly less likely to participate in non-obligatory delivery room tours. Thus, our results might not apply to the general population and may be biased in unknown ways. Third, the women in our sample were likely to have higher levels of knowledge of lifestyle-related risks, as the women were recruited at non-obligatory delivery room tours. Thus, in the general population, pregnant women’s knowledge of lifestyle-related risks during pregnancy is likely to be substantially lower than in our sample. Fourth, we used a non-validated, self-developed questionnaire. However, we pilot-tested the questionnaire carefully with 10 potential participants and then adapted it accordingly. After seven pilot tests, we did not obtain any new information and reached saturation.

Despite the mentioned limitations, we believe that our study provides in-depth insight into pregnant women’s level of knowledge of the lifestyle-related risk factors during pregnancy and their potential health impacts on their offspring, as well as into the factors affecting women’s knowledge of lifestyle-related risk factors during pregnancy. Our preliminary results might therefore be a basis for further research.

## 5. Conclusions

Many pregnant women underestimate lifestyle-related risk factors during pregnancy as compared to available scientific knowledge. Our study emphasized that interventions on lifestyle-related risk factors during pregnancy specifically need to address women from the low to middle socioeconomic status group, and women with statutory health insurance. Interventions on improving pregnant women’s knowledge of health-related risk factors during pregnancy should focus on information on the effects of alcohol consumption, smoking, and a vegan diet. Moreover, interventions should include general information on nutrition and supplementation during pregnancy. In the setting of the study (Germany), providing information in gynecological care settings via face-to-face interactions between pregnant women and health professionals may be an efficient way to improve pregnant women’s knowledge of lifestyle-related risk factors during pregnancy. Future research might benefit from including more women from lower income and education groups.

## Figures and Tables

**Table 1 ijerph-17-08626-t001:** Characteristics of the study population.

	% (*N*)/Mean (SD)
Maternal age	31.7 (4.6)
Week of pregnancy	28.9 (6.7)
Number of previous pregnancies	1.3 (0.6)
Number of biological children	0.2 (0.5)
Level of education	
Low	20.6
Middle	25.0
High	54.4
Household net income	
Low	21.8
Middle	24.9
High	53.3
Marital status	
In a partnership	99.0
Not in a partnership	1.0
Insurance status	
Statutory health insurance	82.2
Private health insurance	17.8

Note: Values are percentages for categorial values and means with standard deviations in parentheses for continuous variables. Data were missing for relationship status (*n* = 2), level of education (*n* = 5), insurance status (*n* = 1), age (*n* = 2), household net income (*n* = 12), and number of pregnancies (*n* = 3).

**Table 2 ijerph-17-08626-t002:** Pregnant women’s knowledge of lifestyle-related risk factors during pregnancy (*N* = 209).

	Alcohol Consumption	Smoking	Coffee Consumption	Recommended Nutrition ^1^	Avoidable Nutrition ^1^	Supplementation ^1^	Medication	Oral Health	Type of Physical Activity ^1^	Intensity of Physical Activity
Consistent with prevailing knowledge (%)	97.1	99.0	37.0	29.2	45.5	6.8	99.5	96.6	51.9	98.5
Inconsistent with recommendations (%)	2.9	1.0	63.0	70.8	54.5	93.2	0.5	3.4	48.1	1.5
Number of false answers (mean (SD))	^a^	A	A	1.03 (0.95)	0.70 (0.95)	1.81 (0.97)	^b^	^a^	0.62 (0.75)	^a^

Note: Values are percentages for categorial values and means with standard deviations in parentheses for continuous variables. Data were missing for alcohol consumption (*n* = 2), smoking (*n* = 2), coffee consumption (*n* = 1), supplementation (*n* = 2), medication (*n* = 2), oral health (*n* = 3), kind of physical activity (*n* = 1), and intensity of physical activity (*n* = 1). ^1^ Answers are based on multiple selection. ^a^ No error calculation due to single selection answers. ^b^ No error calculation possible due to only one false answer.

**Table 3 ijerph-17-08626-t003:** Pregnant women’s knowledge of the health effects of alcohol consumption, smoking, and a vegan diet.

	Effects of Alcohol Consumption ^1^	Effects of Smoking ^1^	Effects of a Vegan Diet ^1^
Consistent with prevailing knowledge (%)	68.2	51.5	17.7
Inconsistent with recommendations (%)	31.8	48.5	82.3
Number of false answers (mean (SD))	0.49 (0.06)	0.90 (1.16)	0.98 (0.70)

Note: Values are percentages for categorial values and means with standard deviations in parentheses for continuous variables. Data were missing for the effects of alcohol consumption (*n* = 14), smoking (*n* = 3), and a vegan diet (*n* = 6). Values are percentages. ^1^ Answers are based on multiple selection.

**Table 4 ijerph-17-08626-t004:** Risks of non-compliant knowledge of lifestyle-related risk factors during pregnancy in association with independent variables, expressed in odds ratios (95% confidence intervals).

	Coffee Consumption (*N* = 190)	Recommended Nutrition ^1^ (*N* = 191)	Avoidable Nutrition ^1^ (*N* = 191)	Supplementation ^1^ (*N* = 189)	Kind of Physical Activity ^1^ (*N* = 190)
Maternal age	1.01 (0.94–1.09)	1.02 (0.94–1.11)	1.05 (0.98–1.14)	0.99 (0.85–1.14)	1.02 (0.95–1.10)
Week of pregnancy	1.00 (0.95–1.05)	0.97 (0.92–1.02)	0.99 (0.95–1.04)	1.06 (0.98–1.15)	1.01 (0.96–1.06)
Number of previous pregnancies	0.96 (0.58–1.59)	1.11 (0.64–1.92)	1.20 (0.70–2.05)	0.94 (0.34–2.64)	1.16 (0.70–1.92)
Level of education					
High	1.00	1.00	1.00	1.00	1.00
Middle	0.42 (0.19–0.90) *	1.13 (0.50–2.57)	2.00 (0.89–4.50)	0.96 (0.22–4.03)	2.78 (1.30–5.98) **
Low	0.88 (0.36–2.15)	1.44 (0.55–3.75)	0.55 (0.22–1.38)	0.75 (0.15–3.68)	1.82 (0.77–4.27)
Level of household net income					
High	1.00	1.00	1.00	1.00	1.00
Middle	1.26 (0.56–2.83)	2.04 (0.85–4.93)	1.68 (0.76–3.70)	0.76 (0.19–3.04)	0.90 (0.41–1.98)
Low	1.01 (0.43–2.41)	1.50 (0.60–3.78)	7.45 (2.59–21.42) ***	3.67 (0.38–35.47)	1.85 (0.78–4.38)
Insurance status					
Private health insurance	1.00	1.00	1.00	1.00	1.00
Statutory health insurance	1.40 (0.60–3.30)	0.86 (0.36–2.06)	1.27 (0.56–2.92)	0.70 (0.14–3.62)	1.54 (0.65–3.60)
Information received by a gynecologist					
Yes	1.00	1.00	1.00	1.00	1.00
No	0.77 (0.35–1.71)	0.94 (0.41–2.18)	1.47 (0.63–3.42)	3.03 (0.37–24.72)	0.84 (0.38–1.87)

Note: Significance key: * *p* < 0.05, ** *p* < 0.01, and *** *p* < 0.001; values are odds ratios with 95% confidence intervals in parentheses. ^1^ Answers are based on multiple selection.

**Table 5 ijerph-17-08626-t005:** Risk of non-compliant knowledge of the health effects of alcohol consumption, smoking, and a vegan diet during pregnancy in association with independent variables, expressed in odds ratios (95% confidence intervals).

	Effects of Alcohol Consumption ^1^ (*N* = 180)	Effects of Smoking ^1^ (*N* = 189)	Effects of Vegan Diet ^1^ (*N* = 186)
Maternal age	1.00 (0.92–1.08)	1.03 (0.96–1.11)	1.00 (0.90–1.11)
Week of pregnancy	1.03 (0.98–1.09)	1.00 (0.96–1.04)	0.96 (0.90–1.03)
Number of previous pregnancies	1.01 (0.58–1.78)	0.85 (0.51–1.42)	1.11 (0.56–2.20)
Level of education			
High	1.00	1.00	1.00
Middle	1.39 (0.60–3.24)	0.57 (0.26–1.22)	1.24 (0.43–3.57)
Low	1.39 (0.53–3.56)	0.59 (0.25–1.38)	0.86 (0.27–2.74)
Level of household net income			
High	1.00	1.00	1.00
Middle	1.64 (0.67–4.00)	1.00 (0.46–2.17)	2.15 (0.73–6.31)
Low	2.82 (1.14–7.03) *	2.31 (0.98–5.46)	3.93 (1.00–15.37) *
Insurance status			
Private health insurance	1.00	1.00	1.00
Statutory health insurance	5.06 (1.11–22.98) *	1.06 (0.47–2.40)	0.45 (0.14–1.46)
Information received by a gynecologist			
Yes	1.00	1.00	1.00
No	2.44 (1.02–5.85) *	1.76 (0.80–3.86)	1.24 (0.42–3.63)

Note: Significance key: * *p* < 0.05; values are odds ratios with 95% confidence intervals in parentheses. ^1^ Answers are based on multiple selection.

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
