# Peer review of "Health Knowledge of Lifestyle-Related Risks during Pregnancy: A Cross-Sectional Study of Pregnant Women in Germany"

_ijerph, 2020, doi:10.3390/ijerph17228626_

Round 1
Reviewer 1 Report
This study is a cross-sectional study investigating knowledge of lifestyle factors during pregnancy and their impact on offspring. The study identified knowledge gaps in women that were associated with economic status and suggested that gynaecologists would be well placed to help any public health promotion messages in this setting.
While I can see the intention of this study, I don’t think it adds significant value to the field mainly because while the authors noted that understanding of the risk factors during pregnancy that may impact the offspring fall under the umbrella term of health literacy, the study itself doesn’t aim to investigate the health literacy status of individuals. Instead, it only identifies risk factors associate with literacy in general rather than health literacy specifically. I also question the method of survey validation as pilot testing on such a small sample seems inadequate.
Also the associations identified between socioeconomic factors and knowledge of recommendations or impact of failing to adhere to guidelines doesn’t appear very consistent or logical. For example, I don’t believe there is a rational explanation as to why health insurance would impact knowledge of the impacts of alcohol consumption but have no bearing on knowledge regarding the impact of smoking. In general, the authors did not adequately consider these inconsistencies nor make compelling arguments for their observations in the discussion. The discussion needs work to make effective arguments to explain these data.
Reviewer 2 Report
This is a very interesting study on the knowledge of pregnant Germans on health knowledge. Overall it is very interesting, the background itself is an excellent quick review of the appropriate literature. I just have a few questions and comments that I believe need to be addressed.
Minor:
Methods page 14, line 158. Please define legal age as this may vary by country.
Methods page 14 line 170. I recommend making the questionnaire available in an online format. A supplementary file might work or an academic depository where it can be easily accessed without contacting the authors. For multiple choice questions this is especially important as the options make things easier or more difficult for those surveyed.
Methods: Dependent variables. Can you define what low/medium/high mean for education level and income?
(Table 3 but also discussion) Vegan lifestyle: How many of the women surveyed were vegan? Or what would you expect? If there aren't many vegans than it is understandable that there is little known. I would consider at least discussing this as it probably the category with the most wrong answers.
Results: Lines 298-301. Is there a reason this isn't in one of the tables? It seems that all of the other data is. Being one of the larger differences (compliance and info from gynecologist) it seems to get lost here.
Reviewer 3 Report
The design of the study is interesting, I would like to see the questionnaire provided for the participants.
The authors stated that the high income and high education level groups were overrepresented. However, it would be of great interest if they included information about the representation of those groups in general German population, as it would allow to evaluate the real extent of study bias.
Furthermore, the authors stated that they included only women who were able to fill in the questionnaire in German - what is the percentage of illiterate and non-German-speaking women in the population analyzed?
What is a "Christian hospital"? Is is it a form of private health service? Maybe it would explain the overrepresentation of high income and education level women?
The manuscript would benefit from a check by an English speaker.
In total, this is an interesting study and some of the results are of great concern. Expanding the study group and including more women from lower income and education groups might be of profit.
Round 2
Reviewer 1 Report
Thanks to the authors for addressing the concerns raised in the last review. I now feel that these have been adequately addressed in the manuscript.